# PRELIMINARY THEORETICAL TROUBLESHOOTING IN VARIATIONAL AUTOENCODER

## ABSTRACT

Variational Autoencoder plays an important role in disentangled representation learning. However, it is found facing posterior collapse problem and learning multiple variants in one factor. What would be learned by variational autoencoder(VAE) and what influence the disentanglement of VAE? This paper tries to preliminarily address VAE's intrinsic dimension, real factor, disentanglement and indicator issues theoretically in the idealistic situation and implementation issue practically through noise modeling perspective in the realistic case. On intrinsic dimension issue, due to information conservation, the idealistic VAE learns and only learns intrinsic factor dimension. Besides, suggested by mutual information separation property, the constraint induced by Gaussian prior to the VAE objective encourages the information sparsity in dimension. On disentanglement issue, subsequently, inspired by information conservation theorem the clarification on disentanglement in this paper is made. On real factor issue, due to factor equivalence, the idealistic VAE possibly learns any factor set in the equivalence class. On indicator issue, the behavior of current disentanglement metric is discussed, and several performance indicators regarding the disentanglement and generating influence are subsequently raised to evaluate the performance of VAE model and to supervise the used factors. On implementation issue, the experiments under noise modeling and constraints empirically testify the theoretical analysis and also show their own characteristic in pursuing disentanglement.

## 1 INTRODUCTION

Variational AutoEncoder(VAE)s (Kingma & Welling (2013), Rezende et al. (2014)) have shown their powerful human-like abilities: modelling causal relationship, unsupervisedly extracting disentangled factors/representation (Bengio et al. (2013)) and generating signals with abundant diversities in a "latent-factor-controllable" way. Those capabilities enable the knowledge transferring through shared causes/factors among different tasks/experiences, emphasized as the important human advantages against the current machine by Lake et al. (2016) and compling with the ideal mental imagery mechanism in memory and thinking. Benefitted from those capabilities, VAEs have been widely applied to various applications, including disentangled representations learning of images and time series (Higgins et al. (2016), Kulkarni et al. (2015), Mathieu et al. (2016), (Fabius & van Amersfoort (2014)), few-shot and transfer learning (Rezende et al. (2016), Higgins et al. (2017b), Higgins et al. (2017a)), causal relationships modeling (Louizos et al. (2017)), pixel trajectory predicting (Walker et al. (2016)), joint multi-modal inference learning (Suzuki et al. (2016)), increasing diversity in imitation learning (Wang et al. (2017)), generation with memory (Li et al. (2016)) and etc.

However, the lack of public theoretical study regarding the generating and inference procedure induced by VAEs is tripping the research process:

- **Intrinsic Dimension Issue**: Could the VAE learn the intrinsic number of factors underlying the data?

- **Real Factor Issue**: Could the VAE learn the real generating factors underlying the data or just some fantasies?

- **Disentanglement Issue**: What are the need and range induced by the word "disentanglement"?

- **Indicator Issue**: Could the effectiveness of current disentanglement metric be guaranteed?

All those lacks distillation would make researchers hard to effectively compute through their knowledge from the experiments and incline to make some avoidable arguments and considerations.

This paper will first discuss the properties of the idealistic VAE [1] to target the aforemention issues and then moving to

- **Implementation Issue**: Could the aforemention analysis be instructive in real implementation?

For **Intrinsic Dimension Issue**, the information conservation theorem shows that idealistic VAE learns and only learns the intrinsic factor dimension illustrated in Fig.(1). To **Disentanglement Issue**, the clarification on disentanglement is subsequently made. For **Real Factor Issue**, the factor equivalence properties shows that idealistic VAE possibly learns all factors in equivalent class rather than exactly pre-specified real factors. For **Indicator Issue**, limitations of the current disentanglement metric are analyzed and several new indicators are introduced.

After that, for **Implementation Issue**, we relax the discussion to the case that decoding procedure are not deterministic through noise modeling perspective. The experiments empirically testify that the knowledge derived form the idealistic case could be applied to the realistic sampling case and demonstrate the behaviors of different noise assumptions as well as our indictors.

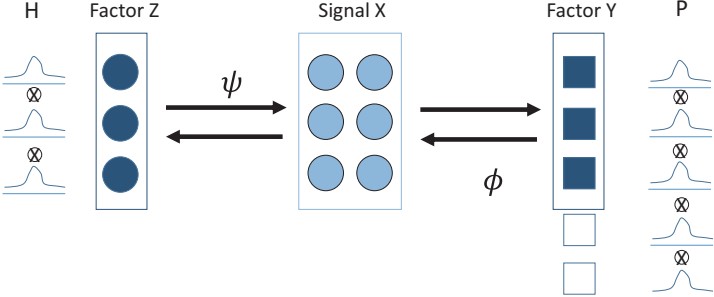

Figure 1: **Idealistic VAE learns and only learns the intrinsic factor dimension.** The illustration of the information conservation theorem 1. Suppose that the oracle data, denoted by random variable $x$, is generated by $y$ (with $P$ independent unit Gaussian random variables) with a homeomorphism mapping $x = \phi(y)$. Idealistic VAE will be forced to learn the factor $z$ (with $H$ independent unit Gaussian random variables) that generates the $x$ with a homeomorphism mapping $x = \psi(z)$. It yields $z = \psi^{-1} \circ \phi(y)$ and $y = \phi^{-1} \circ \psi(z)$. Then according to the information conservation theorem, it must hold that $H = P$.

## 2 GAUSSIAN-VAE MODEL

Gaussian-VAE (Kingma & Welling (2013), Rezende et al. (2014)) is an scalable unsupervised representation learning model (Higgins et al. (2016)), and since Gaussian distribution can be continuously and reversibly mapping to many other distributions, the theoretical analysis on it is also instructive for other continuous latent factors VAE.

Gaussian-VAE assume that input $x$ is generated by several independent Gaussian factors $z$, that is $p_\theta(z) = \mathcal{N}(z|0, I_H)$. The generating/decoding process is modeled as $p_\theta(x|z)$ and the inference/encoding process $q_\phi(z|x)$ is treated as the approximate posterior distribution. Both of them are parameterized by the neural network with parameter $\theta$ and $\phi$.

---

[1]That is, roughly, the VAE whose decoding procedure and encoding procedure are both deterministic.

In VAE setting, the approximate inference method is applied to maximizing the variational lower bound of $p_\theta(x) = \int p_\theta(x|z)p_\theta(z)dz$,

$$\mathcal{L}(q) = \mathop{\mathbb{E}}_{z \sim q_\phi(z|x)} \log p_\theta(x|z) - D_{KL}(q_\phi(z|x)||p_\theta(z)) \leq \log p_\theta(x). \tag{1}$$

## 2.1 IDEALISTIC VAE

In order to get assess to aforemention issues, we will start the analysis from the idealistic situation: *An idealistic VAE model means that it can perfectly encode the signal into "used" factors and perfectly decode the "used" factors to original input signal and the factors follows i.i.d unit Gaussian distribution.*

The idealistic VAE discussed in this literature should also under the following the **Deterministic Assumption on** $q_\phi(x|z)$ **and** $p_\theta(z|x)$. If the factors of $x$ are well understood, the generation process should be deterministic, that is $p_\theta(x|z) = \delta(x = \psi(z))$. We limit the consideration that $q_\phi(z|x) = \delta(z = \psi^{-1}(x))$ is also deterministic as well for simplicity of analysis in this paper. For more complex situation, this consideration could be also basic and instructive.

We try to address aforemention issues by disregarding the training procedure and direct considering the idealistic VAE's behavior.

## 3 ON INTRINSIC DIMENSION ISSUE

In order to get asses to the intrinsic dimension issue, we will present the information conservation theorem. It states some basic truths, e.g. two independent Gaussian and three independent Gaussian cannot be the generating factor of each other under continuous mapping. The theorem thus further illustrates that idealistic VAE learns and only learns the intrinsic factor dimension.

From the perspective of VAE objective, we will also show that the constraint induced by Gaussian prior, plays the lasso on the mutual information which encourage to clip down the small information dimension and promotes information sparsity in factors. In order to derive this perspective, the mutual information separation theorem and objective decomposition theorem are subsequently raised.

### 3.1 INFORMATION CONSERVATION

**Theorem 1** (Information Conservation). *Suppose that* $z = (z_1, \cdots, z_H)$ *and* $y = (y_1, \cdots, y_P)$ *are sets of* $H$ *and* $P$ *($H \neq P$) independent unit Gaussian random variables, respectively, then these two sets of random variables can not be the generating factor of each other. That is, there are no continuous functions* $f : \mathbb{R}^H \to \mathbb{R}^P$ *and* $g : \mathbb{R}^P \to \mathbb{R}^H$ *such that*

$$z = g(y) \quad and \quad y = f(z).$$

Proof in Appendix B. The principle of the theorem is visually illustrated in Fig. 1.

### 3.2 SEPARATION OF THE MUTUAL INFORMATION

The mutual information regarding the factors learned by the inference/encoder network and the signal $x$ can be a good quantity for evaluating the generating influence. That is,

$$\mathcal{I}_{encoder}(x; z) = \mathop{\mathbb{E}}_{x \sim p_{data}(x)} D_{KL}(q_\phi(z|x)||q_\phi(z)). \tag{2}$$

In order to understand and estimate which factor of the VAE was learnt and influenced the generating process, $\mathcal{I}_{encoder}(x; z_h)$ can be taken as a rational indicator[2]. If we assume that $z_1, z_2, \cdots, z_H$ is conditional independent given $x$[3], it can yield a useful result as the following.

---

[2]If $\mathcal{I}_{encoder}(x; z_h) = 0$, it yields $x$ and $z_h$ are independent with each other. The bigger $\mathcal{I}_{encoder}(x; z_h)$, the more information $z_h$ conveys regarding $x$.

[3]It follows the real implementation assumption that $\Sigma_{z|x}(x) = diag(\sigma_{z1}(x), \cdots, \sigma_{zH}(x))$.

**Theorem 2** (Mutual Information Separation). *Let $z_1, \cdots, z_H$ be independent unit Gaussian distribution, and $z_1, z_2, \cdots, z_H$ be conditional independent given $x$. Then*

$$\mathcal{I}(z_1, \cdots, z_H; x) = \sum_{h=1}^{H} \mathcal{I}(z_h; x). \tag{3}$$

Proof in Appendix B. This theorem suggests that if the learnt $q_\phi(z)$ can factorize and the $q_\phi(z|x)$ can factorize, then the consideration of each $\mathcal{I}_{encoder}(z_h; x)$ won't be excess or lose information. Besides, when those term comes in the optimization objective, then it can start the negotiation between the information preservation and dimension reduction and play the role of lasso that clip down the factor in dimension with small mutual information.

**Theorem 3** (Objective Decomposition). *The terminology follows the aforemention definitions and if the involved KL-divergence and mutual information is well defined, then*

$$\mathbb{E}_{x \sim p_{data}(x)} D_{KL}(q_\phi(z|x)||p_\theta(z)) = \mathcal{I}_{encoder}(x; z) + D_{KL}(q_\phi(z)||p_\theta(z)). \tag{4}$$

Proof in Appendix B. The theorem demonstrates that the second term in variation lower bound in Eq. (11) is capable of controlling both the mutual information of $x$ and $z$ induced by the encoder network as well as the similarity of the learnt $q_\phi(z)$ and the prior $p_\theta(z)$. Further, the theorem suggests that it possesses the lasso capacity of clipping down the non-intrinsic factor dimension to some extent, visually demonstrated in Fig.(2).

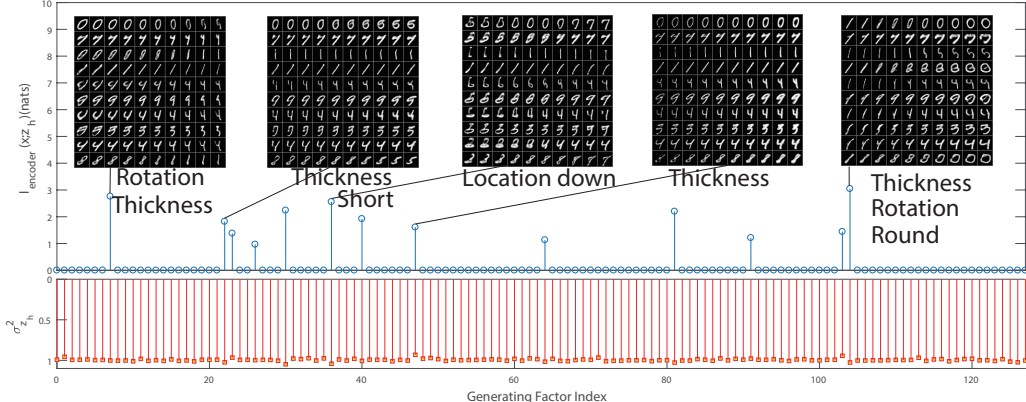

Figure 2: **The sparsity of mutual information occurs; $\tilde{I}_{encoder}(x; z_h)$ determines the "used" factors; disentangled VAE pursues the intrinsic factors dimensions; generating factor exists equivalence class.** Noise learning $\beta$-VAE ($\beta = 10$, equivalent $\sigma^2 = 0.112$):$\tilde{I}_{encoder}(x; z_h)$, $\sigma^2_{z_h}$ and qualitatively influential factor traversals. The top pulse subgraph: $\tilde{I}_{encoder}(x; z_h)$ of each factor. The bottom reverse pulse subgraph: the estimated variance $\sigma^2_{z_h}$ of each factor. The montages: influential factor traversals. We select those factor traversals with visually most interpretable/comprehensive effects to present and the whole influential factor traversals are listed in appendix 13. The phenomenon of the multiple semantic change induced by the same learnt factor and the encoding of same semantic among different learnt factor tallies with factor equivalence class theorem 4. The similar plot of its counterpart with specified normalized noise can be found in Fig.(16) in Appendix.

# 4 ON DISENTANGLEMENT ISSUE

**Inspiration on Disentanglement:** According to the information conservation theorem 1, the independent unit Gaussian factor assumption forms a strong inductive bias and facilitates the model incline to achieve most efficient coding. Under this assumption, the number of the learnt "used" factors of idealistic VAE should be the same as the true factors number under some assumptions such as the learnt $q_\phi(z)$ should equal $p_\theta(z)$ and decode/encode process is continuous and reversible. Empirically, sometimes, though the number latent of factors sometimes is pre-specified larger, only a

small amount of unit Gaussian variables regarding the factors of VAE have been used while $q_\phi(z|x)$ is close to deterministic. The theorem helps provide an interpretation to explain this phenomenon.

Here, in order to avoid the ambiguity of the terminology of disentanglement in this paper, we make the following clarification.[4]

- The disentanglement of the learnt representation/factors in this literature refers to two parts depicted in Theorem 1:
    - **the factors are closer to be independent with each other,**
    - **the factors incline to be able to generate the oracle signal and to be inferred perfectly from the oracle signal through a continuous procedure/mapping.**
- The "disentanglement" refers to the closeness of the learnt factors to the pre-specified independent factors/concpets that can generate the oracle signal and be perfect inferred through a continuous procedure/mapping such as the independent semantic/visual factors.

Therefore, the estimation for $D_{KL}(q_\phi(z)||p_\theta(z))$ that reflects the divergence of the learnt factor distribution and the i.i.d. unit Gaussian prior can be good a indicator to supervise the independence of the factors and served to quantitatively assess the disentanglement of each extracted factor, while the similarity regarding the original signal and reconstruction place another part of the disentanglement.

The "disentanglement" will be shown hard to be obtained in an unsupervised manner. Concretely, even in the idealistic cases, the extracted factors tend to possess the intrinsic number of latent factors of the model, while there are still possibly large variations of these factors due to it can be obtained only in the equivalent class induced by the pre-specified factors as proved in the next section.

## 5 ON REAL FACTOR ISSUE

As for real factor issue, Gaussian Factor Equivalence theorem, (i.e. linear orthogonal transformation of Gaussian factor set are still gaussian factor set.), and Linear Factor Equivalence Class will be presented. They states that idealistic VAE are possibly learns any factors set in the factors equivalence class, and we should not expect "one-to-one" correspondence by disentanglement.

### 5.1 FACTORS EQUIVALENCE

**Theorem 4** (Gaussian Factor Equivalence). *Suppose that $z = (z_1, \cdots, z_H)$ is a set of $H$ independent unit Gaussian random variables. Let $Q \in \mathbb{R}^{H \times H}$ be an orthogonal matrix and then $y = Qz$ is also a set of $H$ independent unit Gaussian random variables. Besides, $z$ and $y$ can generate each other through a linear homeomorphism mapping.*

Proof in Appendix B. This theorem implies that there are a class of unit Gaussian random variables which can generate each other and have equivalent conservation information, as indicated by the following theorem.

**Theorem 5** (Linear Gaussian Factor Equivalence Class).

$$[z] = \{y|y = Qz, \quad Q \in \mathbb{R}^{H \times H} be\ the\ orthogonal\ mapping.\}$$

*Then $\forall y \in [z]$, $y$ is a set of $H$ independent unit Gaussian random variables and can generate $z$ through an linear homeomorphism mapping.*

The theorem clarifies that if Gaussian-VAEs have an linear matrix multiplication freedom degree of learning the factors, then the factors in the equivalence class can all be possibly learnt.

The empirically results tally with the above analysis(see Fig. 3). Suppose the visual semantic concepts can be viewed as a set of independent Gaussian variables ($z =$

---

[4]Notice this clarification is based on the assumption that $q_\phi(z|x)$ is deterministic. If not, then continuous and reversible mapping of encoder constrain should be loosen and also the reversibility of the decoder should be loosen. If we further demand the enhancement of the pattern separation and completion ability, that is, to make the hyperspheres induced by the observation points in the factor domain fully fill up the whole compact factor ball, then some auxiliary constrains including the restriction on mutual information $\mathcal{I}_{encoder}(x; z)$ (defined in Section 3.2) induced by the encoder network need to be introduced.

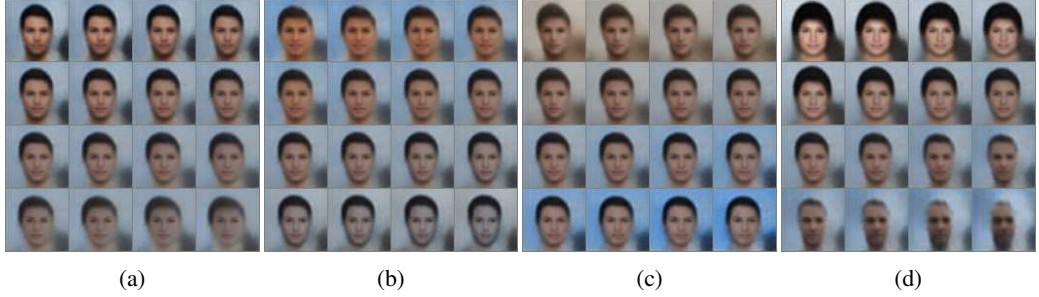

|     |     |     |     |
| :-: | :-: | :-: | :-: |
| (a) | (b) | (c) | (d) |

Figure 3: **We should not expect "one-to-one" correspondence by disentanglement.** One-shot traversal & generating factor equivalence class demonstration. The images are generated by MoG-2 $\beta$(=40) VAE trained on CelebA. The seed image is obtained out from the datasets. Each block represents the traversal of the generating factor from $[-3 + z_{seedh}, +3 + z_{seedh}]$. (a) corresponds to face color white-yellow & female-male change. (b) corresponds to face color white to yellow change. (c) corresponds to background yellow to blue change. (d) corresponds to hair color white to black & face width change. It can be seen that changing one factor results in multiple semantic factor changes in a comprehensible manner rather than the "one-to-one" correspondence which reflexed analysis regarding generating factor equivalence.

$(z_{rotation}, z_{gender}, z_{with-glass}, \cdots)^T)$ which are desired to be captured and learnt by VAEs, while the model is also possible to learn the independent factor set $y = (y_1, \cdots)^T = Qz$ in the equivalence class $[z]$. This explains why changing one factor like $y_1$ sometimes empirically results in change in multiple visual concepts.

This perspective suggests that it's actually hard to obtain the "disentangled representation" that exactly "one-to-one" corresponds to the "independent semantic representation" even though they are in the same equivalence class. As a result, the idealistic VAE model just tend to learn the "entangle representation" if we do preset "oracle generating factors" belonging to the equivalent class.

However, though those conclusions might be upsetting, it seems not be biology impossible. A neuron in hippocampus of animals was suggested to combinatorially possess several representation capabilities. E.g., Aronov et al. (2017) found that some neurons in rat's hippocampus involved in spatial representation also were involved in representing sound frequencies after training rats by a tasks that required them to use a joystick to manipulate sound in frequency continuously.

# 6 ON INDICATOR ISSUE

## 6.1 LIMITATION OF THE EXISTING DISENTANGLEMENT METIC

Higgins et al. (2016) proposed a "simulated factor" based "disentanglement" metric on the simulation datasets. However, according to Gaussian factor equivalence theorem 4 that even idealistic VAE will still learn the factors in the equivalence class, their metric could be effective sometimes for disentanglement but might suffer instability when evaluating the VAE in different trials (detailed in Appendix C).

Further, this metric could be hardly calculated in the real datasets to provide direct feedback of the "disentanglement". The reason is that it must pre-know the generating factors expected to be learnt.

## 6.2 PROPOSED INDICATORS

In order to quantify the disentanglement performance[5] as well as the $\mathcal{I}_{encoder}(x; z)$, we assume that $q^*(z)$ is a factorized zero mean Gaussian estimation for $q_\phi(z)$.

---

[5]Visual recognition could also provide a way to get assess to the factors equivalent class and supervise the disentanglement. If we assume that the most of the visual concept/factors follow the assumption regarding the "disentanglement", it is rational to qualitatively measure the interpretability of extracted latent factors by human perception to infer the disentanglement. Besides, previous empirical evidences of VAE applications

We can then list the indicators for assessing latent factor disentanglement:

**Definition 1** (Estimation for $\mathbb{E}_{x \sim p_{data}(x)} D_{KL}(q_\phi(z|x)||p_\theta(z))$)**.**

$$\tilde{D}_{KL}(q_\phi(z|x)||p_\theta(z)) = \frac{1}{M} \sum_{m=1}^{M} D_{KL}(q_\phi(z|x_m)||p_\theta(z)). \tag{5}$$

**Definition 2** (Estimation for $I_{encoder}(x; z)$)**.**

$$\tilde{I}_{encoder}(x; z) = \frac{1}{M} \sum_{m=1}^{M} D_{KL}(q_\phi(z|x_m)||q^*(z)). \tag{6}$$

**Definition 3** (Estimation for $I_{encoder}(x; z_h)$ which quantifies the influence of each factor)**.**

$$\tilde{I}_{encoder}(x; z_h) = \frac{1}{M} \sum_{m=1}^{M} D_{KL}(q_\phi(z_h|x_m)||q^*(z_h)). \tag{7}$$

**Definition 4** (Estimation for $D_{KL}(q_\phi(z)||p_\theta(z))$)**.**

$$\tilde{D}_{KL}(q_\phi(z)||p_\theta(z)) = \tilde{D}_{KL}(q_\phi(z|x)||p_\theta(z)) - \tilde{I}_{encoder}(x; z). \tag{8}$$

Note that the above indicators 2-4 need the value of $q^*(z)$, we now introduce how to calculate this term based on Theorem 3. Through the minimization equivalence, we know that

$$\min_{Q} \mathbb{E}_{x \sim p_{data}(x)} D_{KL}(q_\phi(z|x)||Q(z)) \Leftrightarrow \min_{Q} D_{KL}(q_\phi(z)||Q(z))dz, \tag{9}$$

the $q^*(z)$ can then be obtained by gradient method from solving the following optimization problem.

$$q^*(z) = \arg \min_{Q} \frac{1}{M} \sum_{m=1}^{M} D_{KL}(q_\phi(z|x_m)||Q(z)). \tag{10}$$

# 7 IMPLEMENTATION ISSUE

## 7.1 VAE WITH NOISE MODELING

In order to testify the idealistic consideration in real situation, we are not going to learn all the factors or equivalently we assume that datasets have noise[6](this situation correspond to that $p_\theta(x|z)$ is not deterministic since the major factors $z$ only forms a subset of the whole factors.[7]), we integrate the noise modeling into our model:

$$p_\theta(x|z) = \mathcal{N}(x|G(z), \sigma^2 I_d),$$

where $\sigma^2$ is either manually enumerated or adaptive learned. Noise modeling can be found crucial in influencing disentanglement in experiment since it would actually define the factors aimed to be learnt and subsequently influence the learnt intrinsic dimension suggested by information conservation property of VAE.

## 7.2 NOISE MODELLING WITH AUXILIARY CONSTRAINT

**The entangled representation can be caused by the over-large of searching space of $q_\phi(z|x)$.** If the learned $q_\phi(z) = \int q_\phi(z|x)p_{data}(x)dx$ has a big divergence to $p_\theta(z)$, then the VAE model tends to learn the entangle representation as it violates the one part of the disentanglement (clarified in section 4). Actually, in the VAE model, what we want is to search in the space that

---

(Higgins et al. (2016), Higgins et al. (2017b), Larsen et al. (2015), Mathieu et al. (2016)) suggest it an effective way.

[6]Noise can be viewed as the generating factor that we are not interested in.

[7]Notice the indeterministic $p_\theta(x|z)$ could lead to the indeterministic $p_\theta(z|x)$, but since the minor factors are supposed to have less influence on $x$, it could not bother the deduction using the knowledge we derived too much.

$q_\phi(z)$ is possibly similar to $p_\theta(z)$ [8]. By implementing this ideal, we add auxiliary upper bound $\mathbb{E}_{x \sim p_{data}(x)} D_{KL}(q_\phi(z|x)||p_\theta(z))$ (detailed in Theorem 3) of $D_{KL}(q_\phi(z)||p_\theta(z))$ to the original objective. This equivalently leads to the approach of $\beta$-VAE raised by Higgins et al. (2016).

$$\sup_{\phi,\theta} \mathbb{E}_{x \sim p_{data}(x)} \mathcal{L}(q_\phi(z|x)) - (\beta - 1)D_{KL}(q_\phi(z|x)||p_\theta(z))$$
$$= \mathbb{E}_{x \sim p_{data}(x)} \mathbb{E}_{z \sim q_\phi(z|x)} \log p_\theta(x|z) - \beta D_{KL}(q_\phi(z|x)||p_\theta(z)) \tag{11}$$

where $\beta > 1$.

### 7.2.1 RELATION OF VAE AND $\beta$-VAE UNDER GAUSSIAN NOISE ASSUMPTION WHEN $\sigma^2$ IS PRE-SPECIFIED

Equivalent objective of $\sigma^2$ pre-specified Gaussian noise VAE:

$$\mathbb{E}_{z \sim q_\phi(z|x)} \|x - G(z)\|_2^2 - 2\sigma^2 D_{KL}(q_\phi(z|x)||p_\theta(z)).$$

Equivalent objective of $\sigma^2$ pre-specified as $\sigma_{pre}^2$ Gaussian noise $\beta$-VAE:

$$\mathbb{E}_{z \sim q_\phi(z|x)} \|x - G(z)\|_2^2 - 2\beta\sigma_{pre}^2 D_{KL}(q_\phi(z|x)||p_\theta(z)),$$

where we call $\beta\sigma_{pre}^2$ the normalized variance.

It's shown that when the $\sigma^2$ is pre-specified, manually tuning it is the same as manually tuning $\beta$ with a fixed $\sigma_{pre}^2$ (for example $\sigma_{pre}^2 = 1$) under Gaussian noise assumption. This equivalence saves our time for extra experiment studying the behaviors of this two cases and we call those two case noise specified $\beta$-VAE.

## 8 EXPERIMENT

### 8.1 DATASET

MNIST is a database of handwritten digits (Lcun et al. (1998)). CelebA (Liu et al. (2015)) is a large-scale celebfaces attributes datasets and only its images are used in our experiments. More details are in Appendix D. The extensive comparison of Gaussian-noise modeling $\beta$-VAE and Gaussian-noise with specified variance $(\beta)$-VAE is made on MNIST based on our indicators to exploring the disentanglement as well as to testify the theorem and analysis from the idealistic case to the realistic sampling case. The experiments on CelebA will be auxiliary to further support the generating factor equivalence class theorem[9].

By setting $\beta$ as different values, we compare the performance of $\beta$-VAE with and without pre-specified noise on MNIST. We specifically listed the result of $\beta(=1)$-VAE in all cases. More details can be found in Appendix D.2.

#### 8.1.1 ON IMPLEMENTATION ISSUE

- **Noise modeling/specification influence the disentanglement.**

The noise specifications and modeling significantly influence the model evidence quantitatively and reconstruction qualitatively, as clearly shown in Fig. (4).

The noise specifications significantly influence the divergence regarding $q_\phi(z)$ and $p_\theta(z)$ and noise specified and noise learning $\beta$-VAE achieve similar disentanglement quantitatively based on the similar indicator behaviours of $\tilde{D}_{KL}(q_\phi(z)||p_\theta(z))$ in Fig. (5a) and the number of normal variance factors Fig. (5b) in regard to the normalized variance.

---

[8] Jensen Shannon Divergence and other integral probability metric which can be good choice and directly be optimized through a adversarial format (Makhzani et al. (2015)) as well. However, in practice, we were defeated by the unstability of training GAN-like model.

[9] The noise assumption is slightly changed into mixture of gaussian in Appendix A.2 for CelebA.

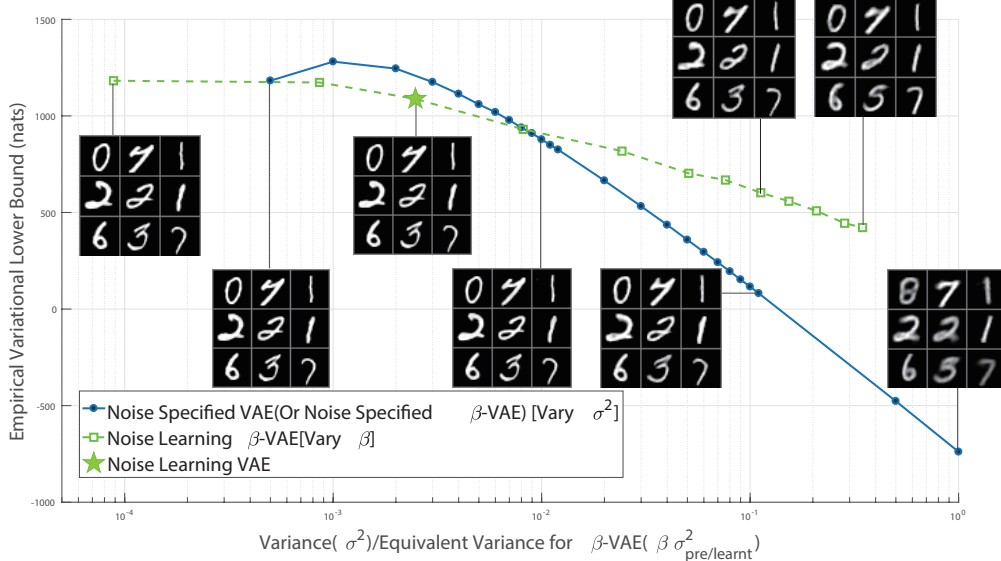

Figure 4: **Noise specification/modeling influence the learned hypothesis and the reconstruction.**
Blue Line: the EVBL (defined in Appendix A.1) of different specified $\sigma^2$ VAE [correspond to pre-specified $\sigma^2_{pre}$ $\beta$-VAE illustrated in Section 7.2.1 with equivalent $\sigma^2 = \beta\sigma^2_{pre}$] . Green Line: the EVLB of noise learning $\beta$-VAE with different specified $\beta$ [normalized to $\sigma^2 = \beta\sigma^2_{learnt}$ for convenient comparing] . Green Pentagram: the EVLB of Gaussian noise learning VAE. Other Figures: their reconstructions on the testing set. The bigger EVLB, the better hypothesis that model learnt.

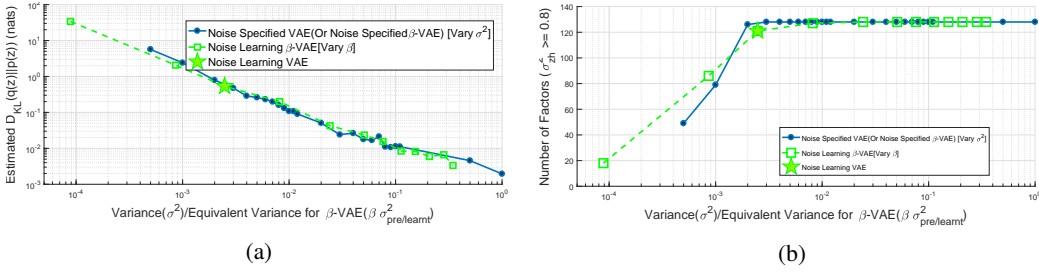

(a)                                                                          (b)

Figure 5: **Noise specification/modeling influence disentanglement.**(a) $\tilde{D}_{KL}(q_\phi(z)||p_\theta(z))$ of different VAE models & (b) Number of normal-variance factors of different VAE models (with 128 factors.)

- **Auxiliary constraints can influence the disentanglement in a different way.**

The prominent difference induced by auxiliary constraints would be its stronger suppression on $\tilde{I}_{encoder}(x;z)$ and the number of influential factors although changing the noise level also possess this capability indirectly. It's somewhat obvious to see and compare value of different indictors under the different $\beta$ setting by sliding on the green/blue line. The bigger $\beta$, the lower $\tilde{D}_{KL}(q_\phi(z)||p_\theta(z))$ and roughly the better reconstruction and hypothesis learnt. However, it's more interesting that in regard to the normalized variance, noise learning $\beta$-VAE enhances the suppression on $\tilde{I}_{encoder}(x;z)$, as depicted in Fig. (6a) and that is comprehensible since $\beta$-VAE is minimizing the auxiliary constraints both $I_{encoder}(x;z) + D_{KL}(q(z)||p(z))$ based on Theorem 3.

### 8.1.2 ON INTRINSIC DIMENSION ISSUE

- $\tilde{I}_{encoder}(x;z_h)$ **effectively determines the "used" factors and VAEs incline to learn the intrinsic factor dimension in realistic sampling case when the disentanglement achieves in some extent.**

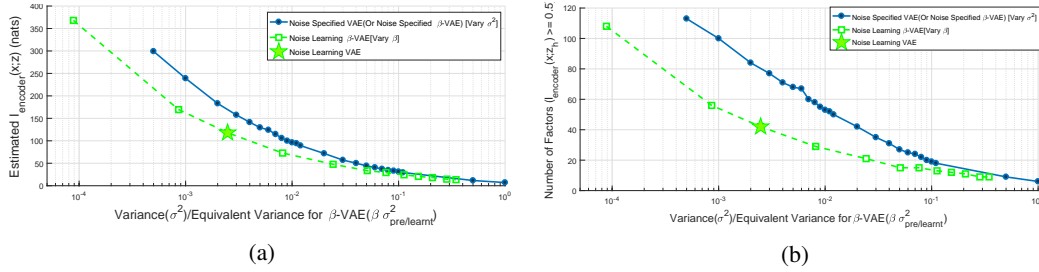

Figure 6: $\beta$-**VAE is more suppressive on the mutual information.** (a) $\tilde{I}_{encoder}(x;z)$ of different VAE models & (b) Number of influential generators of different VAE models

As shown in Fig. (2) and (16), the indicator $\tilde{I}_{encoder}(x;z_h)$ determines the "used" factors. According to those figures, under the suitable noise assumption, VAEs automatically suppress the auxiliary factors and learn the intrinsic factor dimension as it still capable to have good reconstruct abilities and its $D_{KL}(q_\phi(z)||p_\theta(z))$ closer to zero. This phenomenon has already suggested by the information conservation theorem 1.

### 8.1.3 ON REAL FACTOR ISSUE

- **Factor equivalence is generally hold and VAEs possibly learn any factor set in the equivalence class.**

The reflections of the generating equivalence properties 5.1, that is, single factor could result in multiple semantic concepts change and same semantic concept could be encoded in different factors, are again well demonstrated by Fig. (7), Fig. (2), Fig. (16) and Table 1.

Table 1: Variants and factors correspondence on CelebA ( MoG-2 $\beta$(=40)-VAE)

| Variant | glass | height | blue to yellow* | black to white* | half bright half gloomy* |
|---------|-------|--------|-----------------|-----------------|--------------------------|
| Factor  | 73    | 37,45  | 13, 96,40,45,118 | 7              | 110                      |

| Variant | face(big to small) | lighting | face lighting | skin color(white to yellow) |
|---------|--------------------|----------|---------------|-----------------------------|
| Factor  | 63,77,82           | 73,90    | 120,110       | 102,96,28,63,82             |

| Variant | head direction | neck length | hair color | gender | mouth open to close |
|---------|----------------|-------------|------------|--------|---------------------|
| Factor  | 26,31          | 102         | 120        | 28     | 8                   |

* represents background change.

## 9 FUTURE WORK

From the perspective of representation learning:

- It is interesting that the topology properties of oracle signal are used to obtain the proof for the information conservation theorem. Other situations including that data owns several connected components can be further considered and would uncover the efficient coding properties of discrete factors.
- When $q_\phi(z|x)$ is far from deterministic, the discussion would be crucial for many other general purposes induced by word disentanglement. Those study may further extend to the case that data containing different dimension manifolds and to the core VAEs' pattern separation/completion/generalization capabilities.

ACKNOWLEDGMENTS

Thanks for Haodong Sun( Georgia Tech)'s twice greeting me after this paper being rejected.

$-\ _-\ ||$. Here is the acknowledgment to memorize our relationship. ;-)

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

## A  APPENDIX: NOISE MODELING DEDUCTION

### A.1  GAUSSIAN NOISE DEDUCTION

The VAE objective (i.e., the variational lower bound) can be treated as function of the $\theta$ and $\phi$ where noise variance parameters $\sigma^2$ are contained in $\theta$,

$$\mathcal{L}(\theta, \phi, x^m) = \underset{z \sim q_\phi(z|x^m)}{\mathbb{E}} \log p_\theta(x^m|z) - D_{KL}(q_\phi(z|x^m)||p_\theta(z)). \tag{12}$$

Here the SGVB estimator in (Kingma & Welling (2013)), $\tilde{\mathcal{L}}^B(\theta, \phi, x^m) = [\frac{1}{L} \sum_{l=1}^{L} \log p_\theta(x^m|z^l)] - D_{KL}(q_\phi(z|x^m)||p_\theta(z))$ is used. Note that the noise variance $\sigma^2$ is also taken as an optimization variable in the model, making the model capable of better adapting noise variation of data in practical cases in a totally automatic way, instead of a manually set manner.

Given multiple data points from a dataset X, we can construct an estimator of the mean marginal likelihood lower bound of the full dataset, based on minibatches

$$\tilde{\mathcal{L}}^M(\theta, \phi, X^M) = \frac{1}{M} \sum_{m=1}^{M} \tilde{\mathcal{L}}^B(\theta, \phi, x^m), \qquad (13)$$

where the minibatch $X^M = \{x^m\}_{m=1}^{M}$ is a randomly drawn sample set of $M$ datapoints from the full dataset X. Such a lower bound also constitutes an important indicator for model evidence in latter experiment. We call it the empirical variational lower bound (EVLB) in the following.

Note that $\tilde{\mathcal{L}}^B(\theta, \phi, x^m) \simeq \mathcal{L}(\theta, \phi, x^m)$ and we can deduce that

$$\tilde{\mathcal{L}}^M(\theta, \phi, X^M) \simeq \mathop{\mathbb{E}}_{x \sim p_{data}(x)} \mathcal{L}(\theta, \phi, x)$$

$$\leq \mathop{\mathbb{E}}_{x \sim p_{data}(x)} \log p_\theta(x) \leq \mathop{\mathbb{E}}_{x \sim p_{data}(x)} \log p_{data}(x). \qquad (14)$$

The last inequality holds due to $D_{KL}(p_{data}(x)||p_\theta(x)) \geq 0$.

The alternative optimization strategy can be readily utilized to design the algorithm for solving the model by iteratively updating the noise parameter and the network ones. During the optimization process, the objective can be monotonically increasing, and thus the algorithm can be guaranteed to be convergent.

The algorithm is summarized as follows:

**Optimization for parameters except $\sigma^2$:**

**Optimization for $\sigma^2$:** $\sigma^2 = \frac{\sum_{l=1}^{L} \sum_{m=1}^{M} \|x^m - G(z^{m,l})\|_2^2}{dML}$. (Close form solution in regard to $\tilde{\mathcal{L}}^M(\theta, \phi, X^M)$.)

Direct gradient method to the transformed variable $log\_simga = \log \sigma \in \mathbb{R}$ can be implemented to lift the lower bound $\tilde{\mathcal{L}}^M$ as a result to increase the likelihood as well.

### A.2 MoG Noise Deduction

The noise $\varepsilon$ in real situation might be more complex than a simple Gaussian, like that existed in real photographs (Plotz & Roth (2017)). We thus try to further ameliorate the noise setting as a mixture of Gaussian(MoG) noise. Such noise modeling strategy has been widely verified to be effective in applications, like matrix factorization (Meng & Torre (2014)) and robust principal component analysis (Zhao et al. (2014)). That is, we assume that

$$\varepsilon \sim \sum_{k=1}^{K} \pi_k \mathcal{N}(0, \sigma_k^2). \qquad (15)$$

Let $c_d \in \{0, 1\}^K$ be the latent indicator random one-hot variable, $\sum_{k=1}^{K} c_{dk} = 1$, for the MoG-noise component of pixel indexed by $d$. Let $\Pi = [\pi_1, \cdots, \pi_K]$ and $\Sigma = [\sigma_1^2, \cdots, \sigma_K^2]$ be the ratio and variance of each component, respectively. Let $\Pi, \Sigma$ be contained in $\theta$. The conditional joint distribution turns to be

$$p_\theta(c_d, x_d|z) = \prod_{k=1}^{K} \pi_k^{c_{dk}} \mathcal{N}(x_d|G(z)_d, \sigma_k)^{c_{dk}}. \qquad (16)$$

The posterior distribution $q_\phi(z, c|x)$ can be factorized as $q_\phi(z|x)q(c|x, z)$, where $q_\phi(z|x)$ will be direct learnt and the alternative of $q(c|x, z)$, $q(c|x, e)$ will be set to the last step $p_\theta(c|x, e)$ in regard to EM procedure. The lower bound of $\log p_\theta(x)$ is then reformulated as follows:

$$\mathcal{L}(q_\phi(z, c|x)) = \mathop{\mathbb{E}}_{z \sim q_\phi(\tilde{z}|x)} \mathop{\mathbb{E}}_{c \sim q(\tilde{c}|x, \tilde{z}=z)} \log p_\theta(x, c|z) + H(q(c|x, z)) - D_{KL}(q_\phi(z|x)||p_\theta(z)). \quad (17)$$

Similar to the Gaussian case, the reparamerization trick is implemented,

$$\mathcal{L}(q(c|x,e),\phi,\theta,x^m)$$

$$= \underset{e\sim\mathcal{N}(0,1)}{\mathbb{E}} \underset{c\sim q(\tilde{c}|x,e)}{\mathbb{E}} \log p_\theta(x,c|\tilde{z}) + \mathcal{H}(q(c|x,\tilde{z})) - D_{KL}(q_\phi(z|x)||p_\theta(z)), \qquad (18)$$

where $\tilde{z} = En(x) + \Sigma_{z|x}^{1/2}(x)e$.

By utilizing the SGVB estimator, we get,

$$\tilde{\mathcal{L}}^B(q(c|x,e),\phi,\theta,x^m) = [\frac{1}{L}\sum_{l=1}^{L} \underset{c\sim q(\tilde{c}|x^m,e^{(l)})}{\mathbb{E}} \log p_\theta(x^m,c|z^{m,l})$$

$$+\mathcal{H}(q(c|x^m,e^{(l)}))] - D_{KL}(q_\phi(z|x^m)||p_\theta(z)). \qquad (19)$$

Given an input dataset X, we can then construct an estimator to the mean marginal likelihood lower bound of the full dataset, based on minibatches, as follows:

$$\tilde{\mathcal{L}}^M(q(c|x,e),\theta,\phi,X^M) = \frac{1}{M}\sum_{i=1}^{M}\tilde{\mathcal{L}}^B(q(c|x,e),\theta,\phi,x^m), \qquad (20)$$

where $z^{m,l} = En(x^m) + \Sigma_{z|x}^{1/2}(x^m)e^{(l)}$ and the minibatch $X^M = \{x^m\}_{i=1}^{M}$ is a randomly drawn sample of $M$ datapoints from the full dataset X.

Then let

$$p_{\theta_{old}}^{old}(c_d,x_d|z_{old}) = \prod_{k=1}^{K}\pi_k^{old\,c_{dk}}\mathcal{N}(x_d|G^{old}(z_{old})_d,\sigma_k^{old})^{c_{dk}}, \qquad (21)$$

where $z_{old} = En^{old}(x) + \Sigma_{z|x}^{old\,1/2}(x)e$, and we can get

$$p_{\theta_{old}}^{old}(c_d|x,z_{old}) = \frac{p_{\theta_{old}}^{old}(c_d,x_d|z_{old})}{\sum_{c_d}p_{\theta_{old}}^{old}(c_d,x_d|z_{old})}. \qquad (22)$$

The EM algorithm can be naturally employed to solve the model. The implementation steps are listed as follows:

Step 1. **Expectation Step**.

Set $q(c|x^m,e^{(l)}) = p_{\theta_{old}}^{old}(c|x^m,En^{old}(x^m) + \Sigma_{z|x}^{old}(x^m)e^{(l)})$ $i = 1,\cdots,m,l = 1,\cdots,L$.

Calculate the expectation of the latent variable $c$:

$$E(c_{dmlk}) = \gamma_{dmlk} = \frac{\pi_k\mathcal{N}(x_d^m|G(z^{m,l})_d,\sigma_k^2)}{\sum_{l=1}^{L}\sum_{m=1}^{M}\pi_k\mathcal{N}(x_d^m|G(z^{m,l})_d,\sigma_k^2)}, \qquad (23)$$

where $z^{m,l}:z_{old}^{m,l} = En^{old}(x^m) + \Sigma_{z|x}^{old}(x^m)e^{(l)}$.

The Objective in Maximization Step is obtained as the following,

$$\tilde{\mathcal{L}}^M(q(c|x,e),\theta,\phi,x^m) = \frac{1}{M}\sum_{i=1}^{M} -D_{KL}(q_\phi(z|x^m)||p_\theta(z))$$

$$+\frac{1}{L}\sum_{l=1}^{L}\mathcal{H}(q^{old}(c|x^m,e^{(l)})) + \sum_{k=1}^{K}\sum_{d=1}^{D}\gamma_{dmlk}[\frac{(x_d^m - G(z^{m,l})_d)^2}{2\sigma_k^2} + \frac{1}{2}\log(2\pi)\sigma_k^2 + \ln\pi_k]. \qquad (24)$$

Step 2. **Maximization Step:**

Fix: $q(c|x, e)$ determined in the Expectation Step.

$$\frac{1}{M}\sum_{i=1}^{M}-D_{KL}(q_\phi(z|x^m)||p_\theta(z))+\frac{1}{L}\sum_{l=1}^{L}\sum_{k=1}^{K}\sum_{d=1}^{D}\gamma_{dmlk}[\frac{(x_d^m - G(z^{m,l})_d)^2}{2\sigma_k^2}+\frac{1}{2}\log(2\pi)\sigma_k^2+\ln\pi_k].$$

(25)

Update $[\Pi, \Sigma]$ and $\{\theta, \phi\}/[\Pi, \Sigma]$ by alternative optimization strategy.

Update $\Pi, \Sigma$: note here $z^{m,l} : z_{old}^{m,l} = En^{old}(x^m) + \Sigma_{z|x}^{old}(x^m)e^{(l)}$, and we can easily get the closed-form updating formula for these parameters:

$$N_k = \sum_{d,m,l}\gamma_{dmlk} \qquad \pi_k = \frac{N_k}{\sum_{k=1}^{K}N_k} \qquad \sigma_k^2 = \frac{1}{N_k}\sum_{d,m,l}\gamma_{dmlk}(x_d^m - G(z_{old}^{m,l})_d)^2.$$

(26)

Update $\{\theta, \phi\}/[\Pi, \Sigma]$: gradient methods with respect to $\{\theta, \phi\}/[\Pi, \Sigma]$. Note here $z^{m,l} = En(x^m)+\Sigma_{z|x}(x^m)e^{(l)}$.

The algorithm can then be summarized as follows:

1. Initialize the coefficient of $\{\theta, \phi\}/[\Pi, \Sigma]$ and the coefficient of noise $\varepsilon$: $\Pi, \Sigma$.
2. Sample $e$ from $\mathcal{N}(0, I_H)$ to obtain $e_1, \cdots, e_M$ [One for each element sample in the mini batch in the next step ($L$ here is set to 1)].
3. Sample a mini batch $X^M$ from $p_{data}(x)$.
4. Implement EM algorithms as aforementioned (approximate inference for $q(c, z|x)$):
   Expectation: calculate $\gamma_{dmk}$.
   Maximization: update $[\Pi, \Sigma]$, Update $\{\theta, \phi\}/[\Pi, \Sigma]$ with gradient methods.
5. Goto 3: Until Trigger End-Criterion.

## B  APPENDIX: PROOF

*Proof.* For theorem 1. Proof by Contradiction. Suppose those two function exist, and we will show that they will be inverse mapping of each other and the homeomorphism mapping of $\mathbb{R}^H$ and $\mathbb{R}^P$. Since $\mathbb{R}^H$ and $\mathbb{R}^P$ have different topology structures ($P \neq H$), the homeomorphism mapping will not exist.

$$z = g(y) = g(f(z)) \,\forall z \in \mathbb{R}^H \Rightarrow g \circ f = I_H$$

$$y = f(z) = f(g(x)) \,\forall y \in \mathbb{R}^P \Rightarrow f \circ g = I_P$$

Since both $f$ and $g$ are continuous, there is a homeomorphism mapping between $\mathbb{R}^H$ and $\mathbb{R}^P$ and it leads to the contradiction. $\square$

*Proof.* For theorem 4. We only need to test the mean and variance of $y$.

$$\mathbb{E}(y) = \mathbb{E}(Qz) = Q\,\mathbb{E}(z) = 0$$

$$Cov(y, y) = QCov(z, z)Q^T = QIQ^T = I$$

Therefore, $y$ is another set of $H$ independent unit Gaussian random variables. Since $x = Q^T y$, $z$ and $y$ can generate each other with an linear homeomorphism mapping. $\square$

*Proof.* For theorem 2,

$$\mathcal{I}(z_1, \cdots, z_H; x) = \int p(z_1, \cdots, z_H, x)\log\frac{p(z_1, \cdots, z_H, x)}{p(z_1, \cdots, z_H)p(x)}dz_1\cdots dz_H dx$$

$$= \int p(z_1, \cdots, z_H, x) \log \frac{\prod_{h=1}^{H} p(z_h|x)}{\prod_{h=1}^{H} p(z_h)} dz_1 \cdots dz_H dx = \sum_{h=1}^{H} \int p(z_h, x) \log \frac{p(z_h|x)}{p(z_h)} dz_h dx$$

$$= \sum_{h=1}^{H} \mathcal{I}(z_h; x).$$

$\square$

*Proof.* For theorem 3.

$$\underset{x \sim p_{data}(x)}{\mathbb{E}} D_{KL}(q_\phi(z|x)||p_\theta(z)) = \int q_\phi(z|x) p_{data}(x) \frac{q_\phi(z|x) p_{data}(x)}{p_\theta(z) p_{data}(x)} dx$$

$$= \int q_\phi(z|x) p_{data}(x) \frac{q_\phi(z|x) p_{data}(x)}{q_\phi(z) p_{data}(x)} \frac{q_\phi(z)}{p_\theta(z)} dx = \mathcal{I}_{encoder}(x; z) + D_{KL}(q_\phi(z)||p_\theta(z)). \quad (27)$$

$\square$

## B.1 Auxiliary Explanations for Indicators

**Corollary 1.** *The terminology follows the aforemention definitions and if the involved KL-divergence and mutual information be well defined then*

$$\underset{x \sim p_{data}(x)}{\mathbb{E}} D_{KL}(q_\phi(z|x)||q^*(z)) = \mathcal{I}_{encoder}(x; z) + D_{KL}(q_\phi(z)||q^*(z)). \quad (28)$$

The proof of corollary 1 is the same as that of theorem 3. This corollary suggests that the estimation in definition 2 provides another upper bound for the capacity of the encoder network. Empirically, this estimation is a much tighter estimation than using the estimation in definition 1.

**Corollary 2.** *The terminology follows the aforemention definitions and if the involved KL-divergence and mutual information be well defined then*

$$\underset{x \sim p_{data}(x)}{\mathbb{E}} D_{KL}(q_\phi(z|x)||p_\theta(z)) - \underset{x \sim p_{data}(x)}{\mathbb{E}} D_{KL}(q_\phi(z|x)||q^*(z))$$

$$= D_{KL}(q_\phi(z)||p_\theta(z)) - D_{KL}(q_\phi(z)||q^*(z)) \le D_{KL}(q_\phi(z)||p_\theta(z)). \quad (29)$$

The corollary is an direct result of theorem 3 and corollary 1. It suggests that the estimation in definition 4 is a lower bound for $D_{KL}(q_\phi(z)||p_\theta(z))$.

**Definition 5** (Another Estimation for $D_{KL}(q_\phi(z)||p_\theta(z))$)**.**

$$\bar{D}_{KL}(q_\phi(z)||p_\theta(z)) = D_{KL}(q^*(z)||p_\theta(z)). \quad (30)$$

Empirically, $\bar{D}_{KL}(q_\phi(z)||p_\theta(z))$ and $\tilde{D}_{KL}(q_\phi(z)||p_\theta(z))$ shown the same estimation results on M-NIST.

**Definition 6** (Another estimation for $I_{encoder}(x; z)$)**.**

$$\bar{I}_{encoder}(x; z) = -D_{KL}(q^*(z)||p_\theta(z)) + \frac{1}{M} \sum_{m=1}^{M} D_{KL}(q_\phi(z|x_m)||p_\theta(z)). \quad (31)$$

**Definition 7** (Another estimation for $I_{encoder}(x; z_h)$ which quantifies the influence of each factor)**.**

$$\bar{I}_{encoder}(x; z_h) = -D_{KL}(q^*(z_h)||p_\theta(z_h)) + \frac{1}{M} \sum_{m=1}^{M} D_{KL}(q_\phi(z_h|x_m)||p_\theta(z_h)). \quad (32)$$

## C    APPENDIX: ANALYSIS ON THE "DISENTANGLEMENT" METRIC RAISED IN $\beta$-VAE (HIGGINS ET AL. (2016))

The terminology inherits those in the $\beta$-VAE paper. The main idea of that "disentanglement" metric is to create a statistic point $z_{diff}$ relevant to the model for each simulated factor respectively and then to use a linear classifier to project the statistic point to the corresponding index of the simulated factor. If the statistic points induced by the model are easy to be separated then the model is thought to learn "disentangled" representation.

Here, we will argue that even for the idealistic VAE model that follows the disentanglement condi­tons 4 could still receive bad score under that performance metric in some situations.

Suppose that the true simulated factors $v$ follows $\mathcal{N}(0, I_H)$. Then the learnt "used"[10] factor $z$ can be in the equivalence class $[v]$ according to theorem 4. Concretely, there exists an orthogonal trans­formation $Q$ such that $z = Qv$.

Suppose that the simulated factors with index $y$ of $v$ is fixed. Suppose $v_{y-fixed}^1$ and $v_{y-fixed}^2$ are two random variable representing the samples from the $y$-fixed $v$. Then the factors inferred by the idealistic VAE turns to be $z^1 = Qv_{y-fixed}^1$ and $z^2 = Qv_{y-fixed}^2$.

In order to calculate the statistic point $z_{diff}(y) = \mathbb{E}\,|z^1 - z^2|$, we first calculate the mean and variance of $(z^1 - z^2)$.

$$\mathbb{E}(z^1 - z^2) = Q\,\mathbb{E}(v_{y-fixed}^1 - v_{y-fixed}^2) = 0 \tag{33}$$

$$Var(z^1-z^2) = Var(Q(v_{y-fixed}^1-v_{y-fixed}^2)) = QCov(v_{y-fixed}^1-v_{y-fixed}^2, v_{y-fixed}^1-v_{y-fixed}^2)Q^T$$
$$= Qdiag(2, \cdots, 2, \underline{0}_y, 2, \cdots, 2)Q^T = 2I - Qdiag(0, \cdots, 0, \underline{2}_y, 0, \cdots, 0)Q^T = 2I - 2q_y q_y^T. \tag{34}$$

Therefore, $z_{diff}(y)$ can be obtained through the diagonal value of $2I - 2q_y q_y^T$. That is,

$$z_{diff}(y) = \mathbb{E}(|z^1 - z^2|) = 2\sqrt{\frac{2}{\pi}}(\sqrt{(1-q_{y1}^2)}, \cdots, \sqrt{(1-q_{yH}^2)})^T. \tag{35}$$

From the above equation, the location of statistic point is unique determined by $(q_{y1}^2, \cdots, q_{yH}^2)$. When $(q_{y1}^2, \cdots, q_{yH}^2)$ is close to the vertex of the unit cubic for each $y$ then all the statistic points turn to be easily separated.

However, from the perspective of the objective, all the orthogonal $Q$s are with the same potential to be learnt. It seems not to be with a small probability that statistic points of different indexes take similar location. For instance, if $H = 2$ and $Q = \begin{pmatrix} \frac{1}{\sqrt{2}} & \frac{1}{\sqrt{2}} \\ -\frac{1}{\sqrt{2}} & \frac{1}{\sqrt{2}} \end{pmatrix}$ then $z_{diff}(1) = z_{diff}(2)$ cannot be separated while the representation still follows the disentanglement conditions.

Among different trials, the $Q$ might contribute to that "disentanglement" metric but also might not. That explains why that metric is unstable.

## D    APPENDIX: EXPERIMENT DETAILS

We set $L$ to 1, and minibatch size $M$ to be 100 in all practices. All the pixels value have been linear normalized in to [0,1].

### D.1    CLARIFICATION ON THE CORRECTION ON RESULTS

We find in the last version that the code on calculating $q^*(z)$ is wrong. Concretely, the objective of the KL-divergence of two Gaussian is incorrect calculated. That influences the estimation of

---

[10] The auxiliary unused factor is innocuous for the subsequent analysis.

$\tilde{I}_{encoder}(x;z)$, $\tilde{I}_{encoder}(x;z_h)$, $\tilde{D}_{KL}(q_\phi(z)||p_\theta(z))$ and $\sigma^2_{z_H}$ but has little influence on the determination of the "used" factors. So we redo all the experiments on MNIST and delete the relevant results regarding those wrong-calculated indictors on CelebA and Extended Yale Face B.

The new experiment results solve many our past confusions due to the wrong experiment. They are

- Why $\sigma^2_{z_h}$ is strongly correlated with $\tilde{I}(x;z_h)$?

- Why $\sigma^2_{z_h}$ of the used factor is always relative small?

- Why can VAE still learn the "disentangled" representation when the learnt $\tilde{D}_{KL}(q_\phi(z)||p_\theta(z))$ is such big?

Now we know that they are actually not the cases. Something better is that we find our experiment results are more close to our theoretical analysis: when guaranteeing the reconstruction quality in a tolerance range, the smaller $\tilde{D}_{KL}(q_\phi(z)||p_\theta(z))$ the less "used" factors are learnt. That coincides with information conservation theorem: the independent unit Gaussian of the factors assumption facilitates the most efficient coding.

## D.2 MNIST

We split randomly 7000 datapoints according to ratio $[0.6 : 0.2 : 0.2]$ into training set, validation set (no use), testing set. All the indicators and $q^*(z)$ are evaluated/calculated on 10000 datapoints belonging to the testing set. All the seed images used to infer latent code and to draw the traversal come from the testing set.

In all figures of latent code traversal each block corresponds to the traversal of a single latent variable while keeping others fixed to either their inferred ( $\beta$-VAE, VAE). Each row represents a different seed image used to infer the latent values in the VAE-based models. $\beta$-VAE and VAE traversal is over the $[-3, 3]$ range.

The assumed variance $\sigma^2$ of noise specified Gaussian of VAE models is enumerated from $[0.0005, 0.001 : 0.001 : 0.012, 0.02 : 0.01 : 0.11]$. The $\beta$ setting for the noise learning $\beta$-VAE is enumerated from $[0.1, 0.5, 1, 2 : 2 : 18]$.

## D.3 EXTENDED YALE FACE B (GEORGHIADES ET AL. (2001), LEE ET AL. (2005))

We split randomly 2424 datapoints according to ratio $[0.8 : 0.1 : 0.1]$ into training set, validation set (no use), testing set. The model is training on the training set. All the seed images used to infer latent code and to draw the traversal come from the 100 datapoints from the testing set.

In all figures of latent code traversal each block corresponds to the traversal of a single latent variable while keeping others fixed to either their inferred ( $\beta$-VAE, VAE). Each row represents a different seed image used to infer the latent values in the VAE-based models.

$\beta$-VAE and VAE traversal is over the $[-3, 3]$ range.

The $\beta$ setting for the noise learning $\beta$-VAE is enumerated from $[1, 40, 80, 120, 160]$.

## D.4 CELEBA

We split randomly roughly 200000 datapoints according to ratio $[0.8 : 0.1 : 0.1]$ into training set, validation set (no use), testing set. The model is training on the training set. All the indicators and $q^*(z)$ are evaluated/calculated on 10000 datapoints selected from testing set. All the seed images used to infer latent code and to draw the traversal come from the 100 datapoints from the testing set.

In all figures of latent code traversal each block corresponds to the traversal of a single latent variable while keeping others fixed to either their inferred ( $\beta$-VAE, VAE). Each row represents a different seed image used to infer the latent values in the VAE-based models.

$\beta$-VAE and VAE traversal is over the $[-3, 3]$ range.

The $\beta$ setting for the noise learning $\beta$-VAE is enumerated from $[1, 30, 40]$.

## D.5 NETWORK STRUCTURE

| Dataset | Optimiser | Architecture | |
|---|---|---|---|
| Mnist | Adam $1e-3$ Epoch 200 | Input Encoder Latents Decoder | 28x28x1 Conv 32x4x4,32x4x4 (stride 2). FC 256. ReLU activation. 128 FC 256. Linear. Deconv reverse of encoder. ReLU activation. |
| CelebA | Adam $1e-4$ Epoch 20 | Input Encoder Latents Decoder | 64x64x3 Conv 32x4x4,32x4x4,64x4x4,64x4x4 (stride 2). FC 256. ReLU activation. 128/32 FC 256. Linear. Deconv reverse of encoder. ReLU activation. |
| Extended Yale Face B | Adam $1e-4$ Epoch 2002 | Input Encoder Latents Decoder | 192x168x1 Conv 32x4x4,32x4x4,64x4x4,64x4x4 (stride 2). FC 256. ReLU activation. 128 FC 256. Linear. Deconv reverse of encoder. ReLU activation. |
| Extended Yale Face B (Network Parameterized Noise) | Adam $1e-4$ Epoch 1460 | Input Encoder Latents Decoder | 192x168x1 Conv 32x4x4,32x4x4,64x4x4,64x4x4 (stride 2). FC 256. ReLU activation. 128 FC 256. Linear. Deconv reverse of encoder. ReLU activation. |

## E APPENDIX: AUXILIARY GENERATING PICTURE

Note that only the factors with $\tilde{I}_{encoder}(x; z_h) > 0.5$ are shown.

## F APPENDIX: RELATED WORK ON DISENTANGLEMENT

VAE was proposed by Kingma & Welling (2013) and Rezende et al. (2014) to implement the efficient learning and inference in directed probabilistic models regarding continuous latent variables with intractable posterior distributions and in scalable datasets. They introduced a network inference/recoginition model to represent the approximate posterior distribution and utilized reparameterization trick for stochastic joint optimization of a variational lower bound containing the parameters of both the generative/decoder and inference/recoginition/encoder models.

After being raised, many VAE variations have been proposed to boost VAE's capabilities in generation quality and/or disentanglement of the learned representation. In these methods, multiple efforts were made by improving the generative and inference network structures. Typical works along this line include the convolution/de-convolution structure raised by Kulkarni et al. (2015) and ladder structure raised by Zhao et al. (2017)). Some other works advanced the mechanism under the VAE generation/inference processes. Typical works include the iterative attention generation/inference mechanism raised by Gregor et al. (2015), normalizing flow proposed by Rezende & Mohamed (2015) that enhanced the expressive ability of the approximate posterior and its variants (Kingma et al. (2016)).

Despite the improvement to the VAE itself, some other efforts were made by the ensemble between GAN with VAE. E.g., Larsen et al. (2015) unified GAN and VAE to obtain a better reconstruction and a high-level abstracts visual features embedding. Mathieu et al. (2016) also unified GAN and VAE but put emphasis on disentangling factors of variation. GANs without auxiliary design would learn the data distribution disregarding its noise level though suffer from unstable training and mode collapsing (Salimans et al. (2016)) while VAEs would assume a decomposition of the noise and oracle clean datapoint regarding the noise data with an auxiliary prior on the distribution regarding the factors.

Besides, many efforts were made by regularization on the factor distribution or factor generating effect. E.g., Makhzani et al. (2015) introduced an adversarial loss into the latent space of the autoencoder which in idealistic case could learn any kind factor/lantent distribution including those contributing to the disentangled factors/representation. InfoGAN, raised by Chen et al. (2016), introduced the infomax principle to GAN by adding an auxiliary mutual information regularization which enabled the inference of GANs and led to a better disentangled representation as well.

Recently, there is a new VAE variation is proposed by Higgins et al. (2016) who introduced the $\beta$-VAE framework which enhanced the constraints regarding the KL-divergence of the posterior and prior distribution of VAE and showd a novel disentanglement performance. This method has obtained a better performance as compared with conventional VAE methods, especially on its flexible tuning a compromising a parameter beta between the KL-divergence term and the likelihood term (the variational lower bound).

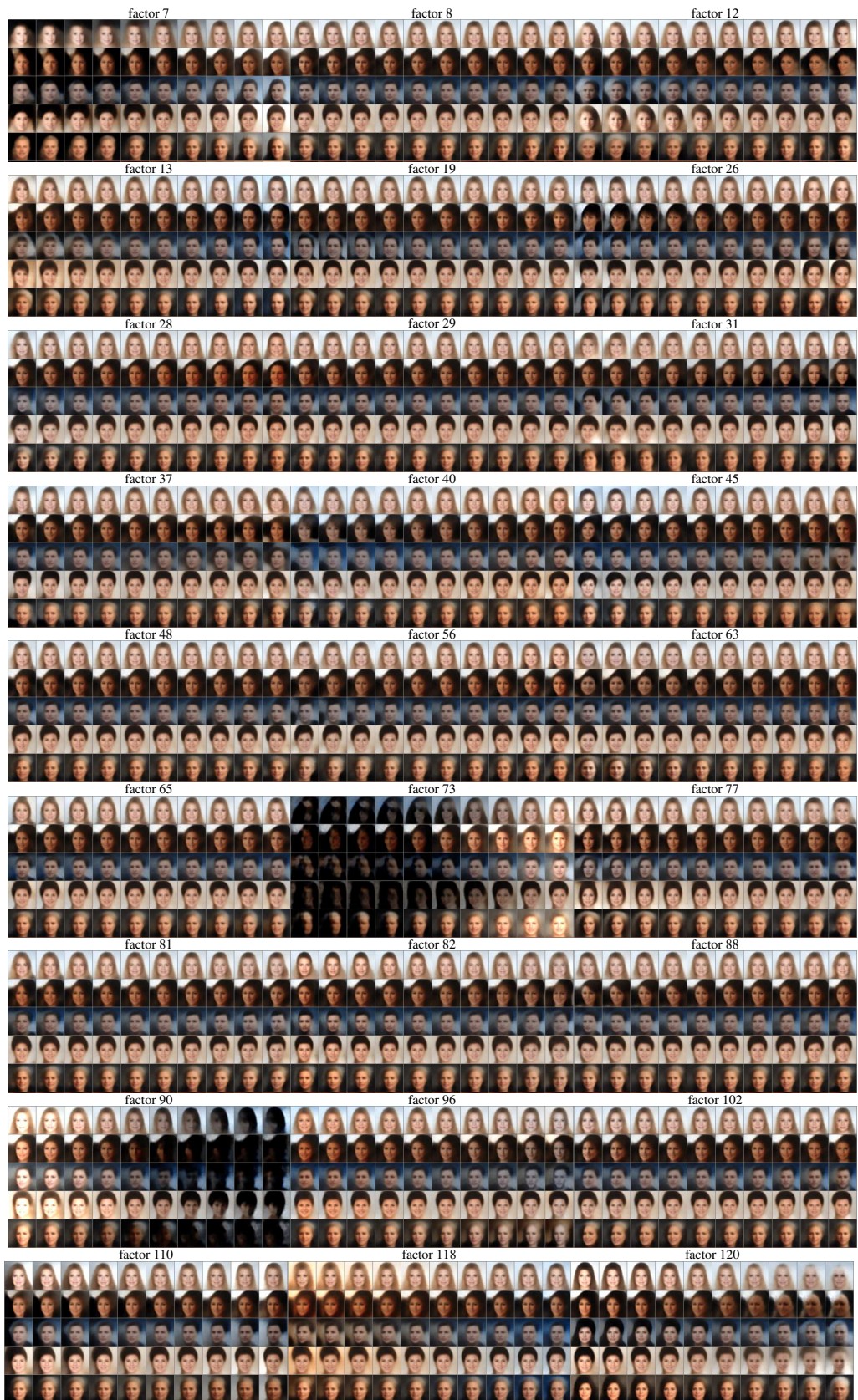

Figure 7: CelebA: Generating Factors Traversal of MoG-2 $\beta$(=40)-VAE.

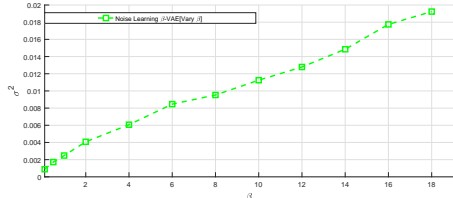 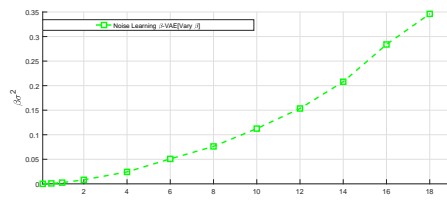

Figure 8: Learned $\sigma^2$ of different $\beta$ setting     Figure 9: $\beta\sigma^2$ of different $\beta$ setting

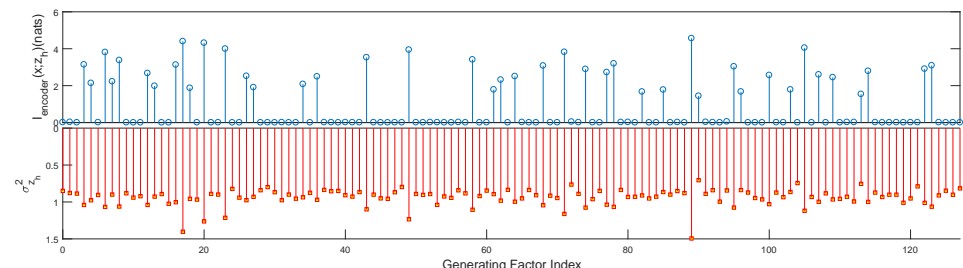

Figure 10: Noise learning $\beta$-VAE ($\beta = 1$, equivalent $\sigma^2 = 0.00248$): estimation of $I_{encoder}(x; z_h)$, $\sigma_h^2$.

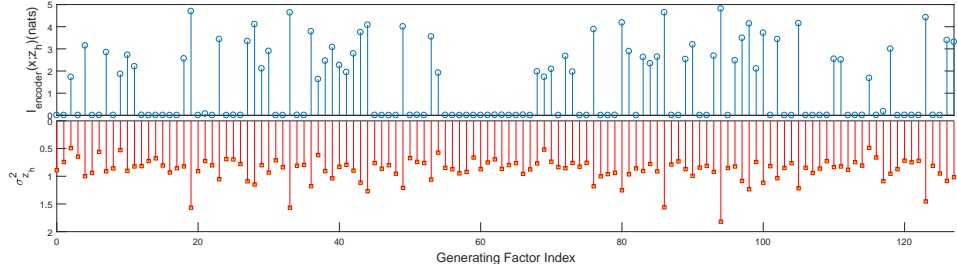

Figure 11: Noise learning $\beta$-VAE ($\beta = 0.5$, equivalent $\sigma^2 = 0.00086$): estimation of $I_{encoder}(x; z_h)$, $\sigma_h^2$.

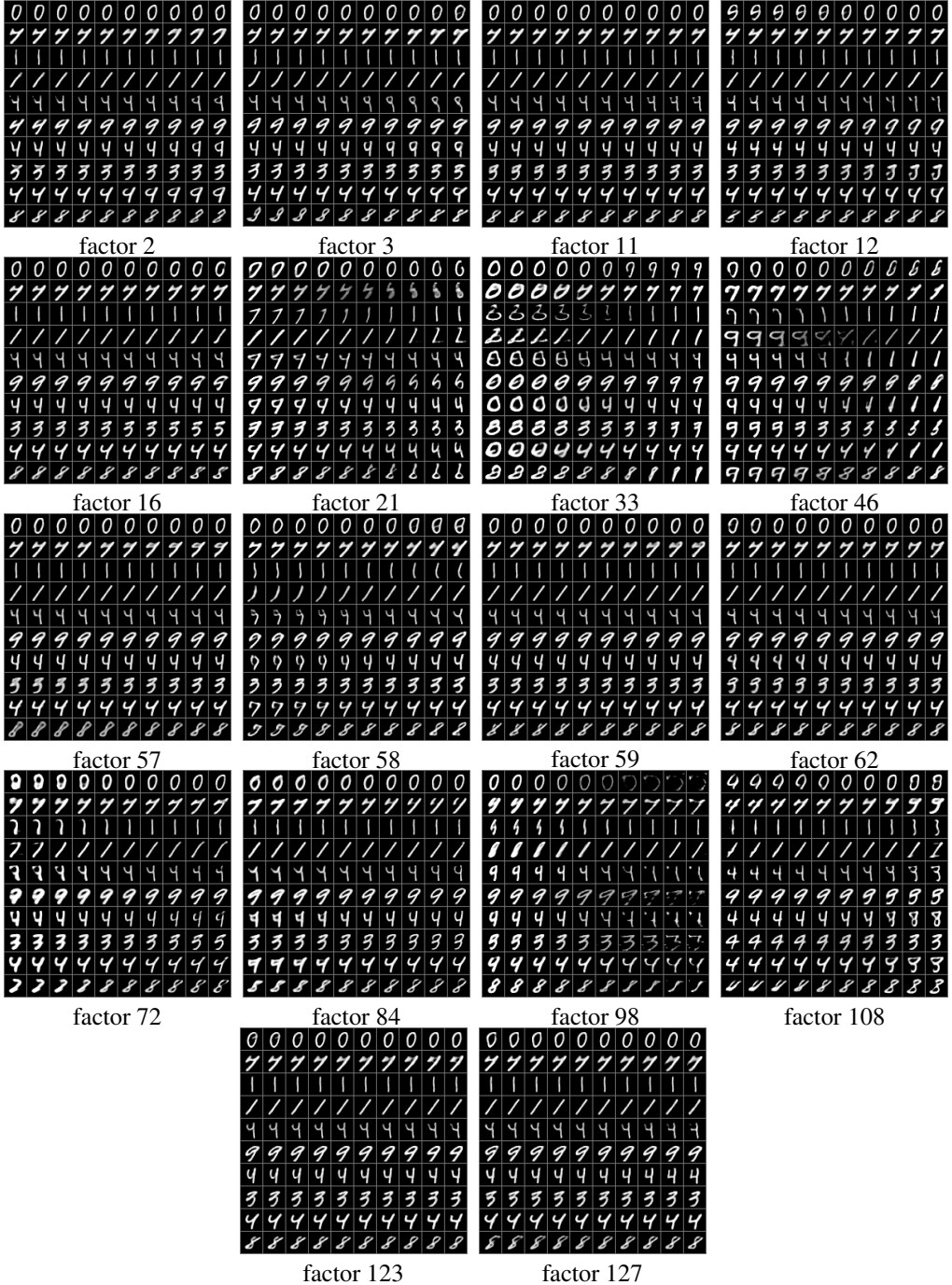

Figure 12: MNIST: Generating Factor Traversal of $\sigma^2 = 0.11$ Pre-specified VAE

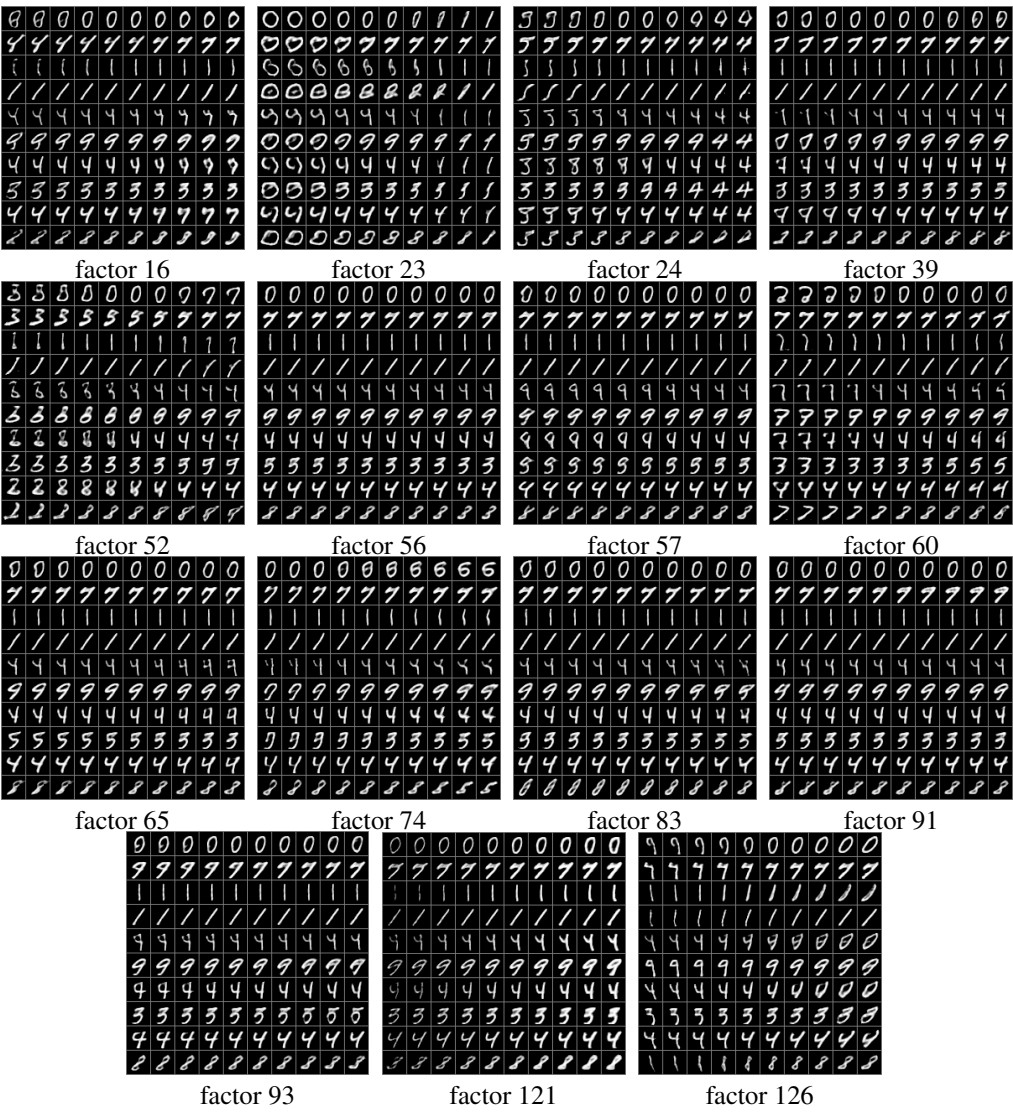

Figure 13: MNIST: Generating Factor Traversal of Noise Learning $\beta$(=10)-VAE

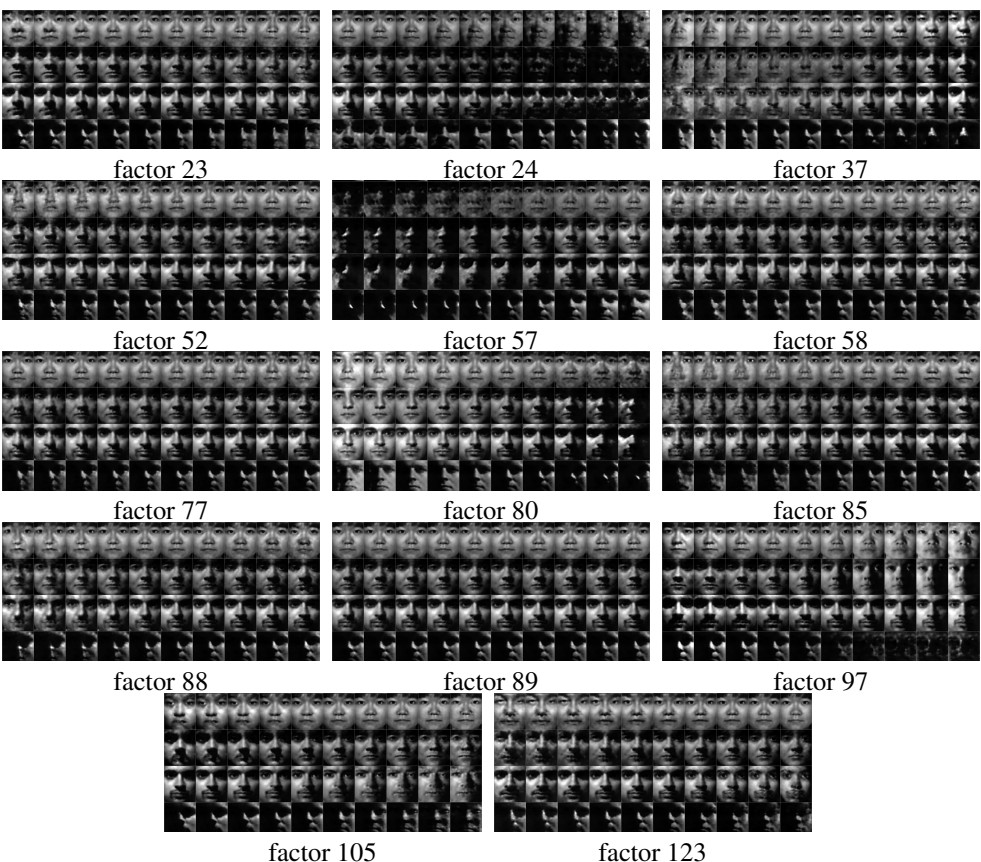

Figure 14: Extended Yale Face B: Generating Factor Traversal of Noise Learning $\beta(= 120)$-VAE . Factor equivalence class properties are still hold.

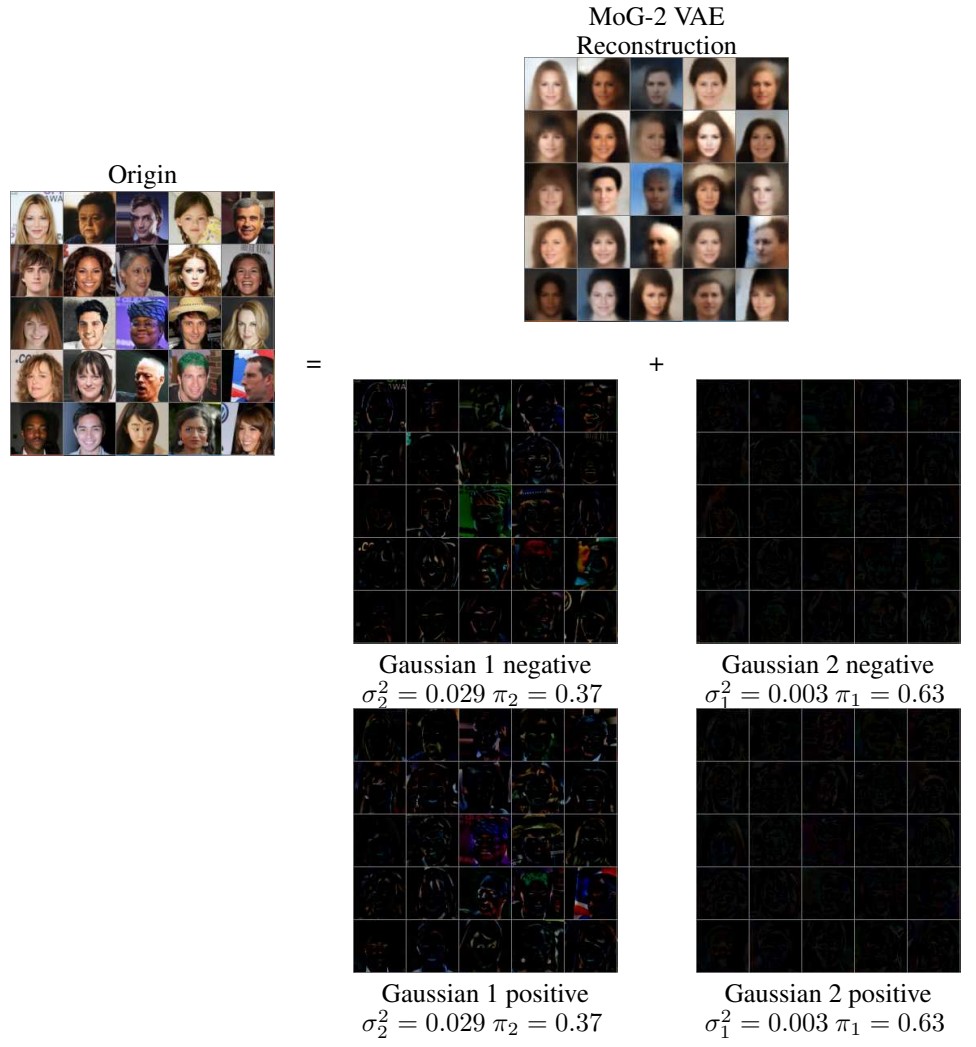

Figure 15: MoG-2 $\beta$(=40)-VAE reconstruction and residual Gaussian components membership

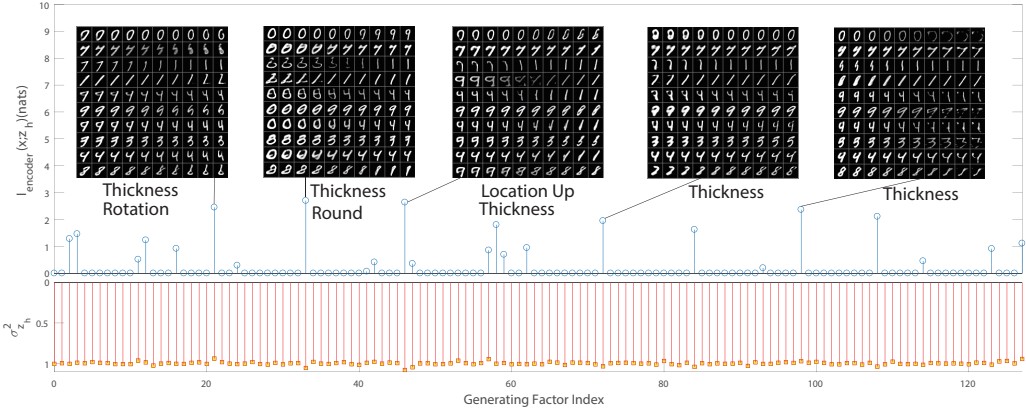

Figure 16: Noise specified ($\beta$-)VAE with equivalent $\sigma^2 = 0.11$: $\tilde{I}_{encoder}(x; z_h)$, $\sigma^2_{z_h}$ and qualitatively influential factor traversals. The mutual information of "used" factor learnt by noise specified $\beta$-VAE can be found more diverse than that in figure 2.

