# OpenReview forum: "Preliminary theoretical troubleshooting in Variational Autoencoder"
_ICLR.cc/2018/Conference — Reject_

### Official Review · AnonReviewer3 · 2017-11-26
**An important problem has been tackled, but not in a satisfactory direction.**

**Rating:** 5
**Confidence:** 4

**Review:**

This paper studies the importance of the noise modelling in Gaussian VAE. The original Gaussian VAE proposes to use the inference network for the noise that takes latent variables as inputs and outputs the variances, but most of the existing works on Gaussian VAE just use fixed noise probably because the inference network is hard to train. In this paper, instead of using the fixed noise or inference network for the noise, the authors proposed to train the noise using Empirical-Bayes like fashion. The algorithm to train noise level for the single Gaussian decoder and mixture of Gaussian decoder is presented, and the experiments show that fitting the noise actually improves the ELBO and enhances the ability to disentangle latent factors.

I appreciate the importance of noise modeling, but not sure if the presented algorithm is a right way to do it. The proposed algorithm assumes the Gaussian likelihood with homoscedastic noise, but this is not the case for many real-world data (MNIST and Color images are usually modelled with Bernoulli likelihood). The update equations for noises rely on the simple model structure, and this may not hold for the arbitrary complex likelihood (or implicit likelihood case). In my personal opinion, making the inference network for the noise to be trainable would be more principled way of solving the problem.

The paper is too long (30 pages) and dense, so it is very hard to read and understand the whole stuff. Remember that the ‘recommended’ page limit is 8 pages. The proposed algorithm was not compared to the generative models other than the basic VAE or beta-VAE.

---

> ### Author Response · Authors · 2017-12-18
> **Response to AnonReviewer3**
>
> We’d like to thank the reviewer for affirmation of noise modeling and their reviews.
>
> 1.	We do agree that using the implicit generative model such as GANs might be a promising way to learn the noise factors.  However, our work focus on the basic framework of VAE and its representation learning properties and capabilities theoretically and practically through noise modeling. Personally, other generative models currently might not be as scalable as Gaussian Prior VAE under proper noise modeling in learning disentangled representation although we do not exclude the possibility of GAN and other implicit generative models could succeed in this subfield in the future. For example, though we did not implement the adversarial loss in the pixel reconstruction domain, we did implement the adversarial loss in the latent space to encourage the disentanglement( they are useful), however, we find even under the same hyperparameter setting their performance can be found really unstable. In contrast, several benefits of VAE are its stability of training model under maximum likelihood principle and the natural inference/encoder capability. By the way, our theoretical analysis on VAE could also be transferable and instructive to the analysis of some other generative models but those theoretical studies and relevant comparison might be left for the future work.
>
> 2.	We personally believe our noise modeling can be extended to many real-world data and could be better than the Bernoulli likelihood modeling. Admittedly, there could be plenty ways people perceive and imagine the noise and data. We observe that you can view the [0,1] range as the pixel-wise probability but still inherits the "cross-entropy" similarity metric to take place of  "E_{q(z|x)}p(x|z)" for "E_{q(z|x)}cross-entropy(x||x_z)". However, this implementation more or less ruins the maximum likelihood principle and the variational procedure, and therefore this implementation disables the usage and intuition of probability knowledge to some extent. In particular, if you want to further discuss the noise on the probability mass, everything could be awkward. As the result, it may need a new theory to enable the knowledge to be accumulated if those assumptions or optimizations are undertaken.
>
> 3.	We condense the original paper into 10 pages (*excluding* references and appendices). The appendices are necessary to enable the reader to assess our proofs and details of experiments and algorithms.  The discussion on the noise modeling is weakened due to the length limitation and emphasis is put on the theoretical discussion of the VAE properties. However, readers can still access the algorithm of MoG-noise VAE in appendices.

---

### Official Review · AnonReviewer1 · 2017-11-27
**Because of the excessive length, poor presentation quality, and limited novelty, this paper is below the bar for ICLR at this time.**

**Rating:** 3
**Confidence:** 4

**Review:**

This paper proposes to modify how noise factors are treated when developing VAE models.  For example, the original VAE work from (Kingma and Welling, 2013) applies a deep network to learn a diagonal approximation to the covariance on the decoder side.  Subsequent follow-up papers have often simplified this covariance to sigma^2*I, where sigma^2 is assumed to be known or manually tuned.  In contrast, this submission suggests either treating sigma^2 as a trainable parameter, or else introducing a more flexible zero-mean mixture-of-Gaussians (MoG) model for the decoder noise.  These modeling adaptations are then analyzed using various performance indicators and empirical studies.

The primary issues I have with this work are threefold:  (i) The paper is not suitably organized/condensed for an ICLR submission, (ii) the presentation quality is quite low, to the extent that clarity and proper understanding are jeopardized, and (iii) the novelty is limited.  Consequently my overall impression is that this work is not yet ready for acceptance to ICLR.

First, regarding the organization, this submission is 19 pages long (*excluding* references and appendices), despite the clear suggestion in the call for papers to limit the length to 8 pages: "There is no strict limit on paper length. However, we strongly recommend keeping the paper at 8 pages, plus 1 page for the references and as many pages as needed in an appendix section (all in a single pdf). The appropriateness of using additional pages over the recommended length will be judged by reviewers."  In the present submission, the first 8+ pages contain minimal new material, just various background topics and modified VAE update rules to account for learning noise parameters via basic EM algorithm techniques.  There is almost no novelty here.  In my mind, this type of well-known content is in no way appropriate justification for such a long paper submission, and it is unreasonable to expect reviewers to wade through it all during a short review cycle.

Secondly, the presentation quality is simply too low for acceptance at a top-tier international conference (e.g., it is full of strange sentences like "Such amelioration facilitates the VAE capable of always reducing the artificial intervention due to more proper guiding of noise learning."  While I am sympathetic to the difficulties of technical writing, and realize that at times sufficiently good ideas can transcend local grammatical hiccups, my feeling is that, at least for now, another serious pass of editing is seriously needed.  This is especially true given that it can be challenging to digest so many pages of text if the presentation is not relatively smooth.

Third and finally, I do not feel that there is sufficient novelty to overcome the issues already raised above.  Simply adapting the VAE decoder noise factors via either a trainable noise parameter or an MoG model represents an incremental contribution as similar techniques are exceedingly common.  Of course, the paper also invents some new evaluation metrics and then applies them on benchmark datasets, but this content only appears much later in the paper (well after the soft 8 page limit) and I admittedly did not read it all carefully.  But on a superficial level, I do not believe these contributions are sufficient to salvage the paper (although I remain open to hearing arguments to the contrary).

---

> ### Author Response · Authors · 2017-12-18
> **Response to AnonReviewer1**
>
> We’d like to thank the reviewer for their making effort to reviewing and providing helpful suggestions although they didn't provide fair assessments of our contribution, especially the important content which appears later that used to reveal some basic facts and behaviors of idealistic VAE as well as our indicators.  We have made a number of changes to address them.
>
> A.	We condense the original paper into 10 pages. We also try to reduce the number of strange sentences.
>
> B.	We weaken our discussion on noise modeling due to the limitation of the paper length and strengthen the theoretical troubleshooting  of   VAE's properties  and they are listed below
>
>     1.	Intrinsic dimension Issue:  "Could the VAE learn the intrinsic number of factors underlying the data?
> Our paper: Yes, idealistic VAE learns and only learns the intrinsic factor dimension and the VAE objective induced by the Gaussian prior also encourages the information sparsity in dimension which is contributing to the learn the intrinsic dimension.
> Besides, in real implementations, the conclusion is also instructive if the noise is proper modeling and the disentanglement(clarified in our paper) is achieved to some extent.
>
>     2.	Disentanglement Issue:  "What are need and range induced by word disentanglement?"
> We provide the clarification according to information conservation theorem:
> the learned the factors are close to being independent.
> the factors incline to generate the oracle signal and to be inferred perfectly from the oracle signal through a continuous procedure/mapping.
>
>    3.	Real Factor Issue:  "Could the VAE learn the real generating factor underlying the data or just some fantasies?"
> We show that idealistic VAE possibly learn any factors set in the equivalence class. Besides, the experiment results also suggest that the VAE's factor equivalence generally exist.
>
>    4.	Indicator Issue: "Could the effectiveness of current disentanglement metric be guaranteed?"
> We show that the current disentanglement introduced by (beta-VAE) is based on "simulated factors" while idealistic VAE possibly learns any factor set in equivalence class induced by the "simulated factors". Hence, that metric may work sometimes and suffer instability among different trials.
> We further introduce some indicator regarding the mutual information I(x;z) and Dkl(q(z)||p(z)) which provide the assessment to the determination of ``used factors" and to the disentanglement.
>
>    5.	Implementation Issue: "Could the aforementioned analysis be instructive in real implementation?'
> We introduce noise modeling to relax the consideration of the real situation. The experiment results empirically testify the knowledge derived from the idealistic case could be instructive in the real situation.  They also demonstrate own characteristic of noise modeling in pursuing the disentanglement.
>
> C.	Despite the theoretical discussion on the intrinsic properties of VAE, if we just discuss the novelty of noise modeling of VAE alone, we don't think it is limited.  If you find different noise assumptions/specifications just significantly influence the disentanglement you will believe it.

---

### Official Review · AnonReviewer2 · 2017-11-27
**Substantial work needed**

**Rating:** 2
**Confidence:** 4

**Review:**

This paper attempts to improve the beta-VAE (Higgins et al, 2017) by removing the trade-off between the quality of disentanglement in the latent representation and the quality of the reconstruction. The authors suggest doing so by explicitly modelling the noise of the reconstructed image Gaussian p(x|z). The authors assume that VAEs typically model the data using a Guassian distribution with a fixed noise. This, however, is not the case. Since the authors are trying to address a problem that does not actually exist, I am not sure what the contributions of the paper are.

Apart from the major issue outlined above, the paper also makes other errors. For example, it suggests using D_KL(q(z)||p(z)) as a measure of disentanglement, with lower values being indicative of better disentanglement. This, however, is incorrect, since one can have tiny D_KL by encoding all the information into a single latent z_i. Such a representation would be highly entangled while still satisfying all of the conditions the authors propose for a disentangled representation.

Given the points outlined above and the fact that the paper is hard to read and is excessively long, I do not believe it should be accepted.

---

> ### Author Response · Authors · 2017-12-06
> **Cannot encode all the information into a single continuous latent z_i without considering the topology structure of data and properties of encoding mapping.**
>
> This comment is an illustration of our perspective and a quick response to the reviewers’ counterexample in their second paragraph comment. We are trying to show that counterexample doesn’t exist in the idealistic situation.
>
> Example:
> Suppose the input X follows a 2 dimension independent unit Gaussian. Let us say we want to encode X into only one dimension unit Gaussian latent Z and decode it back to X.
>
> Discussion:
> To simplify the situation, we could first consider the idealistic encoding and decoding procedure that the q(z|x) =\delta(z=f(x)) and p(x|z)=\delta(x=g(z)) are the two deterministic procedures. [correpsonds to no-information-loss channel case]
>
> Then we have z=f(g(z)) for all z in R^1 ,and x=g(f(x)) for all x in R^2.
>
> If we further assume, the encoder f and decoder g are both continuous mappings (It's innocuous since the continuity of a mapping is a weak condition and mappings induced by the neural network are typically continuous.), then f and g are Homeomorphism mappings. However, R^1 and R^2 spaces have different topological structure and there thus is no Homeomorphism mapping between those two spaces. This leads to the contradiction.
>
> Conclusion:
> In short, roughly, in the idealistic situation, there is no way to encode 2 dim Gaussian into 1 dim Gaussian and then perfectly decode them back through continuous procedures.
>
> Relation to our paper:
> The aforementioned argument can also be found in the information conservation theorem in Intrinsic Dimension Issue(section 3) in the latest paper.

---

> ### Author Response · Authors · 2017-12-18
> **Response to AnonReviewer2**
>
> We thanks the reviewer for their work. However, we're afraid they may have misunderstood the point of our paper and didn't provide fair assessments of our contribution. We hope our responses below and the comments of the other reviewers may help clarify the scope of our work and its significance.
>
> According to your suggestion that paper is too long, we amend the length of our paper from 19 pages to 10. In order to achieve this and keeping it still comprehensive and informative, we have to weaken the discussion on noise modeling and put more emphasis on the intrinsic properties of VAE. We hope our amendment can increase the information channel capacity between the proposed ideal and our readers and provide better and more friendly reading experience.
>
> A.	We understand your first consideration that there exist some implementations under other noise assumptions including Bernoulli distribution for two-point valued data and some other more sophisticated ones with specific oriented domain knowledge. Some of them might already enable the parameter of noise to be learned. However, also in many implementations on real-valued data, many papers just simplified Gaussian assumption to be sigma^2 I where the sigma^2 is either assumed to be known or to be manually tuned. In particular, the tutorial on VAE(https://arxiv.org/abs/1606.05908), which is a really respectable work and is also my first contact with VAE model, is also under this noise assumptions.  And we personally believe that we are the first one publicly emphasizes and demonstrates on "noise modeling influences the disentanglement" though we have also shown some other benefit could be induced by noise modeling in our original paper version.
>
> B.	We thank for your intuitive counterexample to our clarification on disentanglement.  We have proved several theorems especially the information conservation theorem[e.g. two independent Gaussian and one Gaussian cannot be the generating factor set of each other.] in Intrinsic Dimension issue(section 3 in the latest version) to exclude this counterexample in the idealistic case and our experiment results also turn to support the instruction suggested by the theorem in the real implementation. In order to theoretically illustrate our perspective regarding the counterexample that review proposed, more compactly and informatively, we also add an auxiliary deduction in the latter comment. We will be grateful if the reviewer or someone else can further provide some facts and evidence from the theoretical perspective or experimental perspective to show the existence or the probability of the existence of that counterexample in the real situation.
>
> C.	You also mention there might be some other theoretical errors. We are grateful if you can list them. We are open to the opinion and argument from the other side and believe those arguments can improve the direction of scientific research and accelerate our mission to the AI. This will be helpful to improve our work but also good for the whole community.

---

### Author Response · Authors · 2018-01-05
**Revision**

(1) we condense our paper from 19 pages to 10 pages. The remain contents are the subset of the original one.

(2) Due to the length limitation, we shift emphasis from noise modeling to the discussion on the intrinsic properties of VAE and troubleshooting. In particular, the noise modeling is viewed as a crucial part of implementation issue. The following issues, also illustrated in the original version, become the focus of this revision paper:

  A. Intrinsic dimension Issue:  "Could the VAE learn the intrinsic number of factors underlying the data?
  B. Disentanglement Issue:  "What are need and range induced by word disentanglement?"
  C. Real Factor Issue:  "Could the VAE learn the real generating factor underlying the data or just some fantasies?
  D. Indicator Issue: "Could the effectiveness of current disentanglement metric be guaranteed?"
  E. Implementation Issue: "Could the aforementioned analysis be instructive in real implementation?'

(3) Some original discussions on the noise modeling algorithms and related work were moved into the appendices to guarantee the reader can still get access to the MoG-noise VAE algorithm.

(4)  Some original experiment results, discussing the behaviors of MoG-noise VAE and Network Parameterized Noise VAE, are moved to appendices. The programming on the calculation of indicators is found wrong and we have corrected it and redone the relevant experiments. The details can be found in the appendices.

---

### Decision · Program_Chairs · 2018-01-29
**ICLR 2018 Conference Acceptance Decision**

**Decision:**

Reject

**Comment:**

The reviewers agreed that the paper was too long (more than twice the recommended page limit not counting the appendix) and difficult to follow. They also pointed out that its central idea of learning the noise distribution in a VAE was not novel. While the shortened version uploaded by the authors looks like a step in the right direction, it was not sufficient to convince the reviewers.